# "I Didn't Have the Language": Young People Learning to Challenge Gender-Based Violence through Consumption of Social Media

Lena Ida Molnar 

Social and Global Studies Centre, RMIT University, Melbourne, VIC 3000, Australia; s3688150@student.rmit.edu.au

**Abstract:** In recent years, young people in Australia and abroad have taken to social media to express their concerns about the violent behaviour of their peers, and to share content that challenges the causes of gender-based and interpersonal violence. From launching policy-changing petitions to responding to and engaging with online campaigns, young people are pushing action and momentum from generational changes in feminist movements. Young people have their own contexts and influences that affect understandings and responses to gendered inequality and violence. This paper discusses the findings of nine focus groups with 32 young people who share content online about preventing gender-based violence, exploring their perceptions on their reasonings for using these tools and how they came to assumptions about gender inequality. It explores the contexts that young people in Australia draw upon to challenge existing gender inequalities and their reasonings for using social media to share ideas about preventing violence with others. The findings of this paper, thus, have implications for how young people are engaged in the primary prevention of gender-based violence, suggesting better use for social-media-campaign content engagement.

**Keywords:** focus groups; primary prevention; gender-based violence



## 1. Introduction

Feminist movements in Australia have, since the 1970s, sought to bring to light the need for support and prevention strategies to stop gender-based violence before it can occur [1]. Social media has built collaborations and brought attention and response to gender-based violence [2–4], in what the press has more recently called a "reckoning" [5]. However, many who challenge such power can confirm that those who support existing hierarchies will oppose any movement for change. While public interest calls attention to the necessity for transforming structures through the legal system or representation in power, some [6] highlight that such appeals can often reinstitute the violence of material inequalities against the body via gender, race, class, ability, and sexuality. Indeed, public interest and discussion about gender-based violence in the media often focus on those advocates who appeal to the public for palatable solutions through political commodification rather than structural change [7–9]. For those marginalised by multiple sources of power, such as gender, age, race, ability or sexuality, the conditions that position their challenge to violence also put them at heightened risk of backlash and further struggles.

In Australian policy discourse, young people are targeted as 'agents of change' for primary prevention of violence against women (hereafter PVAW) in the settings of education and home as these areas are most easily identifiable for legislators to engage specific youth cohorts for behaviour change [10,11]. Young people aged 16–25 are typically presented as change-makers as this stage in life is depicted as a time of transition and upholding promises for the future generation [12–14]. However, government policy to engage young people towards change can often miss the mark, dismissing or stigmatising their lived

experiences as they include marginalisation, violence, and active sexuality. Such framings can create odds with the theoretical guidance behind primary prevention of violence against women which requires the involvement of social settings towards an 'ecological' transformation of societal attitudes, ideas, and behaviours that support violence, as they can divide practice into ageist or individualist mechanisms [15,16]. To challenge and transform the gendered nature of violence in every social setting, it is essential to engage each community member to create change from their understanding. Violence prevention experts [17] have more recently argued that such an approach builds upon the audience's strengths towards greater appreciation and adaptation of content toward sustaining actions in their social context.

Frequently, however, the prevention of violent behaviours among young people is not viewed through a gender lens [18]. Furthermore, in some contexts, an overemphasis on physical violence alone can distract from other experiences of gendered harm in peer and romantic relationships [19–22]. For these reasons, nonphysical forms of gender-based violence, such as emotional, psychological, or technology-facilitated abuse and harassment (including, but is not exclusive to, behaviour that perpetrates sexual and gendered-based control, harassment, non-consensual sexting, and image-based abuse using technology [22]), are often misinterpreted or overlooked by young people and those within their institutions. As such, there is little Australian literature that discusses how young people experience and respond to or challenge the broad range of gender-based violence in their social lives [23–27]. While much research focuses on the prevalence of sexual and intimate violence among young people, more research is emerging that focuses on the experiences of gender-based violence within the age cohort of relevance to this doctoral research (between 15 and 25 years), inclusive of peer and familial relationships, as well as public, employment or institutional relationships [26,28,29]. This research exists in a broader context of inequalities contributing towards prevention, including the prevalence of gender-based violence that young people experience.

This paper first draws on research about young people's understandings of gender-based violence, in the areas of the home, education, amongst peer groups and through engagement with media technologies, to reflect on the current understandings informing primary prevention efforts targeting young people. I then describe the methods used in this study, which sought to explore the purposes of young people's use of social technology for sharing content about preventing gender-based violence. In doing so, this paper brings attention to the perspectives of young people themselves in using social media content when looking to understand and challenge the causes of violence from their standpoint. The findings highlight the key areas that these young people notice and challenge in discussing gender inequalities that lead to gender-based violence, and how they came to use social media to share these ideas and messages.

## 2. Young People's Knowledge about Gender-Based Violence

There is a genuine concern that early experiences of observations of family violence and gender inequality can lead to a higher susceptibility to experience or partake in similar behaviour [30,31]. Considerable international and Australian research finds family life as an acute setting where young people learn about gender roles, expectations, and behaviours [30–33]. Literature on gender-based violence finds that the home often reproduces young people's violence-supporting behaviours and attitudes [34,35]. Additionally, the Personal Safety Survey [36] illustrates that respondents who had witnessed violence to a parent of the same gender were twice as likely to experience violence in their own relationships.

Familial constructions can influence how young people feel about addressing the issues they meet or how they judge values in their relationships [37]. Through demonstrations of relationship dynamics and household roles, young people see their worlds to accommodate gendered patterns and manage everyday relationships, including conflict and violence [32,37–39]. Indeed, the NCAS found that 16% of young people in Australia shared the idea that such behaviour should be addressed within the family, rather than

challenged by other people [20]. The attitude amongst young people that domestic or family violence is a private concern impacts the severity of the experience and the potential for others to help or speak out [40]. Even so, research on the influence of families indicates that, while limited, parents with equitable social contexts can enable and support values that challenge typical gender roles [41,42]. Accordingly, the behaviours and expectations set up in the home can both perpetuate and challenge many myths about gender, sex and relationships as young people interpret the patterns. This study provides a unique insight into how some young people in Australia reflect on and challenge their early assumptions about gender and relationships and how they are challenged and supported to prevent gender-based violence collectively.

Education settings are important sites for both learning and the early socialisation of young people's gendered practices. As scholarship into these areas finds, the structural order of primary school can often repeat in informal activities where children might 'correct' the subject position of girls to boys in play [43]. Others argue that there is an urgent need for research and policy in Australia for ongoing resourcing of prevention of sexual harassment within schools, to understand how these practices are used to aggressively exclude those who experience multiple marginalisations [44]. Further research highlights how responses to sexual harassment and violence tend to prioritise boys' comfort to maintain school harmony [45]. Furthermore, the instances of sexualised behaviour at elite schools in Australia and the UK reflect a legacy entrenched in racial and class privileged masculinities [18,45]. In these studies, while girls experiencing violence are described by teachers as "dramatic" or "overreactive", boys' emotions were more likely to be soothed rather than given the opportunity to sit with discomfort [18,45]. However, others point out that while teachers in Australia are positioned to use market-based approaches to frame their practices in a context where gender equity has long sat unfavourably, feminist practices often receive backlash [18]. Thus, opportunity for feminist discussion, or the creation of clubs at schools, by teachers or students, is "no easy road" without support from a school [46].

For young people, some of the most grounding and important ways to understand gender, romance, and sexual experiences are peer relationships or friendships. Friendships enable and cement young people's interpretations and understandings of gender roles, romantic or sexual encounters, and gender-based violence. Many young people value their friends and peers as trustworthy sources of information about sex and relationships over their family members [47]. As friendships and relationships fluctuate over youth development, the broader social context of these relationships helps structure views about their romantic lives [43,48,49]. Peer relationships and friendships are often gendered and thus can reproduce differentiation through the 'rules' of friendship [48]. Between friends, young people endorse one another's knowledge about how each other 'should' behave in their first romances and more generally towards others [43,50]. Hence, it is essential to understand how the rapidly changing contexts of young people's social lives structure their gendered practices.

Studies about friendship groups highlight how dominant masculinities and men's behaviour influence attitudes that support gender inequality and gender-based violence [51–53]. Ethnographic studies of men's friendship groups that many will encourage physical, emotional, or sexual abuse of women in response to challenges to assumed patriarchal rights [54]. Young men with friends who use violence in relationships were more likely to do the same [54]. Similarly, other studies about young heterosexual men using violence in relationships show that they are more likely to have friends who support one another's behaviour for popularity among their peers [52]. Young men in Australian secondary schools are found to often use sexual harassment and aggressive behaviour to marginalise women and sexual minorities to increase their own popularity [43]. These tactics reinforce a gendered order within peer groups and a dominant construction of heteronormativity within friendships. These studies show that young men are likely to develop and reinforce similarly harmful beliefs and practices in a context promoting gender inequality and

supporting patterns of violence [51–54]. Consequently, there is a significant area to be researched concerning how young people can discuss what is helpful to influence challenging violence amongst their peers.

Young people also take to social technologies to respond to their experiences, with the literature illustrating that these discussions continuously develop and enable challenges to everyday misogyny [55–57]. Youth practices of social organising and counter-publics enable alternative resourcing and skill building to challenge rape cultures [4,58,59]. However, such private online spaces do not always give clear examples of responding to or supporting cultural change [60]. Some young people seek out material online that reflects their experiences or the experiences of others that they recognise, as this will speak to them and is easy to find [61–63]. Examples of these have been hashtag campaigns, social media groups and blogs that collectivise experiences of racism and misogyny via activism [64]. Some young people who encountered such content or created their own use the experiences of online spaces to build community [59]. They argue that using specific social media platforms where content has been curated and shared can offset the force of violent structures that are often taken for granted [60]. Indeed, young people are more likely to feel included when their practices of culture, play and resistance are accepted and enacted within responses [13]. This study explores how participants adopt and critically engage with such practices to challenge gender-based violence.

Centring the voices of young people remains a key focus of recommendations emerging from research about the prevalence and response of gender-based violence among youth cohorts [10,26,27,65]. Recent findings from a large-scale national study demonstrate that young people in Australia consciously recognise that gendered socialisation is uneven, but not necessarily how this is structured towards inequity and violence [27]. This understanding of the influence of gender can lead young people to see a depiction of campaign messaging about violence prevention [66], for example, as reinforcing stereotypes about men rather than supporting equality. While young people often acknowledge and are sceptical of the influence of media stereotypes in their everyday life [27,67], they can also find themselves adopting their own typically gendered patterns. These behaviours and ideas are typically picked up in the areas of the home, education, from friends, as well as more often from social media. However, there remains a gap in understanding how young people themselves perceive and challenge these messages.

## 3. Methods

Much of the research about young people's engagement with PVAW campaigns has been quantitative or self-evaluative. More recently, the literature indicates the need to explore how young people use such content and training in their lives [57,65]. While quantitative research can provide marketing analytics for such campaigns in providing a general picture of their impact, qualitative methods are used in this study to describe the practices and understandings of young people who engage with PVAW content [10,57]. Thus far, the qualitative research engaging young people towards the prevention of gender-based violence covers interventionist approaches within campaigns [10,65], rather than those exploring the practices of young people themselves. Conscious of the fact that young people are frequently in a power dynamic where they are spoken 'for' or 'about' and research is done 'to' them, I allowed my participants to set the terms of the focus groups, by asking them to provide the prompts, and selecting the most similar materials to discuss. Adapting participatory methods towards focus groups, the following method hypothesises that young people have their own projects [68], underscoring the standpoint of young people to explore how they come to challenge violence and inequality themselves.

This paper presents findings from nine focus group discussions with 32 young people aged between 16 and 25 years old who have shared content via social media about preventing gender-based violence. These focus groups were conducted throughout 2020 and asked participants to share social media content that they may have liked, shared, or engaged with about gender equality and preventing gender-based violence. The most similar of the

submitted content was screen-shared during focus group discussions via Zoom to provide prompts for discussion about the participant's perceptions of the purpose and influence of using content about gender inequality and preventing gender-based violence in their lives.

Young people were recruited through public social media engagement, with support from local government organisations, women's health services, youth services, and grassroots organisations. Participants lived in metropolitan and regional parts of the Australian Capital Territory (2), New South Wales (3), South Australia (2), Queensland (4), Tasmania (1), Victoria (17), and Western Australia (2). There were 23 young women, 4 young men, and 2 young non-binary people. Two participants did not disclose their gender identity. Participants represent demographics including transgender and queer young people, Aboriginal and Torres Strait Islander, first-generation cohorts from Southeast Asia, Western Asia, and Europe, as well as one young person with a disability. While there were a number of young people who did identify as white or Caucasian, were also a number of young people who did not disclose any cultural background or offer other identity features.

The discussions were semi-structured to cover topics about how the participants came to notice and challenge ideas about gender inequality, and the purpose of using social media to better understand those ideas. This approach allowed participants to ask one another questions, follow tangents, or return to other topics we had discussed earlier. These young people reflect on how realising the manner that patterns of gender inequality that was reproduced and challenged at home, school, and amongst their peer groups, led them to seek further knowledge about preventing violence by reaching out online. This starting point reflects on the current academic knowledge to highlight the importance of young people's use of lived experience to challenge violence in the context of primary prevention, illustrating the understandings that young people might have when engaging with or responding to content about preventing gender-based violence at home

The focus group discussions were recorded, transcribed verbatim and then analysed using reflexive thematic analysis [69]. Thematic analysis is a well-regarded in feminist research practice for its iterative approach toward recognition of both researcher positionality and the standpoint of young people [70,71]. I was also able to draw on insight from my own professional background in violence prevention research and program delivery across Australia, as well as lived experience of sexual violence at an early age. The dataset was first read and coded to identify and confirm the themes across the focus groups, then compared with the topics from the literature, and the findings of the first part of this study [66]. Participants of the research were provided with an executive summary of the findings soon after the first inductive analysis to provide comments. The conceptual approach used to guide this analysis was broadly taken from materialist feminism [72,73]. This approach enables analysis of gendered power structures, their reproduction and potential transformation through discursive norms and embodied practices. Overall, the process of reflexive thematic analysis provided the ability to both refine the meaning of the data through appreciation of researcher subjectivity as well as the broader context of the scholarly literature.

## 4. Results

The findings of this paper illustrate the lived, felt moments that young people reflect on to challenge and find support from others when sharing content online. The key themes described in this paper cover how the settings of home, school, and popular culture provided examples to voice action and change in their own words. Participants in this study said that experiences of gender inequality are taken for granted, but using social technology exposed them to a broader understanding and connection with the actions of others.

### 4.1. "There Wasn't a Word to Describe It": Gender Inequality at Home and in the Community

Focus group participants constructed their family and communities as an interconnected influence for early understandings of gender-based violence and inequality. The

home informs understanding for young people, as well as challenging the cultural values of their parents and observing an unequal balance of power between family members. For a few participants, expectations of familial roles were repeated and honoured as cultural traditions. For other participants, such as Lindsey, noticing inequality "was just, like, throughout the whole lived experience, noticing the different ways that men and women are treated" and realising the prevalence of gender-based violence.

While home life dynamics often reproduced the social patterns of traditional gender roles, they were often, in turn, spaces where the participants were exposed to their loved one's experiences of violence. To illustrate, Merindah shared that she noticed her mother's experiences of discrimination and violence, but only began to understand them in hindsight:

> Merindah: But if I try and think when I first started noticing gender equality, I think it would be looking at the way that my mum was treated by different people. Looking at my mum's own experiences with gender equality.

> Interviewer: So, you noticed that your mum was treated differently than the men in your life or other people?

> Merindah: Yeah, and I've just been beginning to understand my mum's own experiences of violence because she was a woman, but also her own experiences of being disbelieved or invalidated when we were navigating different services, because she was a black woman as well.

Merindah speaks of her mother's first-hand experience of being subjected to violence from men and dismissal by services because of her gender, race, and disability. By saying that she is only "beginning to understand" her mother's experiences, there is a recognition from Merindah that as a young person, she was unable to separate her observations of violence from the minimisation that her mother received socially and through institutions. In the discussions with focus group participants, constructions of gender and practices that inscribe the values of the dominant class were frequently referred to as oppressive and sometimes violent.

Participants who identified as women often connected as they discussed being socialised to notice how other women also internalised inequality through the minimisation of violence as they grew up. Some young women felt that inequality marked their understanding of gender through socialisation. Kendra said: "I feel like growing up as a woman does it to you a little bit. It's a little bit; I don't know; maybe you push it under the rug for years". For participants such as Kendra and Naomi, minimising women's experiences of violence often led to accepting inequality until later in life. In Naomi's words:

> "I know my mum grew up in very significant domestic violence and as a teenager I was, my stepdad was perpetrating [violence] towards myself, and my best friend was going through [something] similar. But it was always kind of weird because it sucked, but I wasn't like "Oh, this could be different". When you grow up with it and you're like "Well it shouldn't be like this, but that's just kind of how it is" until you, sort of, are around, at least for me, around people saying it can be different, and then it kind of changed."

Here, Naomi reflects on her lived experiences of violence in her household being normal to the point that she did not consider challenging it. As for other participants, it was not until later in life that these experiences were given a name. Naomi explicitly mentions the collective power of challenging an assumed norm from peer engagement.

The dynamics in familial relationships enable distinct patterns informing early understandings of inequality regarding men's structural position over women [74]. Unequal responses to conflict and gendered performances between parents have been used as examples by young people to justify gendered behaviour and/or the violence of men against women [75]. When young people normalise men's apparent 'natural' sexual aggression through the heteronormative discourses presented in families, they are more likely to minimise violence [75]. Unequal and dismissive parental relations often play a socially

reproductive role in engaging the construction of a dominant, successful father and a passive, nurturing mother [32,76]. Other participants noted the expression of gender roles through the unequal structural arrangements of their parents. Grace found that this was made explicit by her parents giving preferential treatment to her brother:

> "Mum is always doing the cooking, cleaning, looking after everyone and Dad is someone that goes to work and then comes home, and she's already got dinner on the table ready for him. Even Mum, like, making me help her with the cooking and my brother being upstairs, like, just playing on the PlayStation or whatever, like, a difference in the gender roles in the home, that home, that's probably unequal there."

For participants, expectations of others in the family or community also upheld the expressions of gendered practices at home. For Frances, there were limitations on expressing themselves freely to express typical femininity. For example, they said that "as a kid I really wanted to try out makeup, because it looks fun, but whenever I suggested it-that it was so bad. You could just not suggest this because of what someone might say". Here, Frances illustrates how their parents' understandings of gender and social constructions of masculinity placed restrictions upon play and self-expression. These underlying expectations of roles in the home described by participants illustrate the insidious ways that inequality is reinforced.

Understandings of biological gender roles upholding a 'tradition' of dominant masculinity were a common feature of inequality, and such patterns were of cultural significance in participants' families. While some young people discussed roles as being implicit at home, for others, views of gender were more explicit and seen as having a direct link to their cultural background. Yas, a first-generation Australian, similarly described this disconnect or point of difference between her family's emphasis on duty and responsibility, and the conceptualisations of equality she encountered outside her family. Reflecting on this dissonance, Yas observed that "there wasn't quite a word to describe it, you know, a phrase or something" about the disconnect between her family's values about 'equality' from those of the people around her. These participants' descriptions resonate with a sense of how personal experience challenge what the world may lead one to expect [77]. These expectations, and their fundamental connection to a hierarchical family structure, caused these participants to view equality with a lens that was distinct from that of their peers.

The 'lack of words' that participants mention is relevant for some participants as they come to find and then use messages that challenge their experiences of inequality or violence. This suggests that much of their knowledge and their practices in the face of oppression rest on a shared material basis [78]. For Saavi, a young person of South Asian descent, gender represented an additional 'layer' that impacted how she spoke out amongst her peers. Saavi reflected on how these experiences impacted her experiences of sexism within her community when she also challenged racism:

> "I think for me, growing up as a person of colour, I had a lot of elements of racial discrimination towards me and especially from a really young age. I was always a little bit outspoken, and I think when the whole gender equality stuff kicked in was at that very young age for me when, you know, I would see men in my community that were also racially discriminated against come out and speak out and their voices being much more respected than mine with that added gender layer as well."

For young women of colour, gender was an 'added layer' in the struggle to access equality. In the focus groups, they talked about comparing their home lives with others, speaking out, and noticing that their treatment was different in a way they lacked the words. For other participants, their local school community had words for the underlying norms of sexist behaviour. Many participants found it was through shared connections with others that they could notice and reflect on inequality; it was also from finding they

were at a loss to find a way to connect these experiences with others, and they sought ways to speak out about them, such as connecting with resources or online content.

While many focus group participants noticed a pattern where their parents reproduced heteronormativity and traditional gender roles, a few other participants also spoke of more equal examples at home. For Paloma, having "feministy" parents meant she had a close support network to identify sexual harassment when it happened. She said:

> "My mum's always been, like, a very strong feminist, like, radical within some circles, whatever. My dad is quite 'feministy' as well, which we like. So, from quite a young age I would say, kind of, being catcalled when I was, like, a young teen, I knew who to talk to."

Here, Paloma reflects that being modelled a trustworthy understanding of gender-based violence and feminist thought from a young age gave her the ability to discuss experiences of harassment with her parents. Similarly, some participants reflected that noticing and reflecting on gender was grounded in household discussions where equity was frequently raised. For instance, Brendan said: "I didn't have that constant presence [of men]. I had a single mum, so her presence around me sort of obviously made me view things very differently and more supportive towards women in general". These findings echo studies of people brought up amongst parents and community members with more understanding of gender-based issues and sexual diversity [41,42].

### 4.2. "All That Stupid Little High School Stuff": Regulating Inequality in School Settings

For focus group participants in this study, important moments of recognising and experiencing gender inequality were felt during high school. While schools' structure and replicate the gendered behaviours modelled at home, some participants also found schools provided opportunities to subvert these expectations and challenge experiences of violence with peers. For young people in this study, gendered segregation was part of the infrastructure of schools. Merindah summarised this, recalling that "school was a big institution where we're very clearly categorised in girl-versus-boy a lot of the time". Participants could relate how the expectations of passive femininity and aggressive masculinity were then replicated within the social structure of educational institutions. Experiencing that unequal treatment between boys and girls was a prompt for Lindsey:

> "I guess, treatment of the girls in the school compared to the boys and all that stupid little high school stuff that just made me feel really uncomfortable, and then I started doing more research and learning more."

Here, Lindsey recalls how experiencing differential treatment between peers eventually prompted her to start looking into what was making her feel uncomfortable. These findings reflect research into the structuring of gendered patterns in educational institutions that restrict the expression of students through limitations of gender and the prioritisation of boys [18,79]. For these participants, progressing through school presented them with more affective associations of gender alongside the organisation of activities as structural gender inequalities.

A focus on physical capacity during group activities at school and sports classes strongly reinforced for some participants that there were some things that men and boys were allowed to do that woman could not and should not. For some of the participants, the organising structure of the school relied upon assumptions about gender and one another's capabilities. Zoe shared that she noticed inequality when teachers preferred boys for physical tasks. She said:

> "Yeah, I think, especially when teachers would ask boys to help with lifting heavy things in classrooms and I would always ask to do that, but I would never get picked. So, I think that's probably the first time I noticed it."

Here, Zoe highlights how the structuring of school activities often categorically prioritises involving boys over other students. For Yas, this cut deeply when teachers told her she

was not allowed to play soccer with the boys; this established that there was "something about" her:

> "When I really wanted to play soccer but there weren't any girls' teams, and I wasn't allowed to join the boys' team. And just having that recognition that I was different, like, in some way. There was something about me that meant I couldn't be with them."

Yas describes how this barrier to participating in a team sport affected her understanding of her sense of belonging and value drawn from her gendered capacity. These reflections challenged the repeated practices that defined what boys and girls can do together often reinforcing sex-based constructions of masculinity and femininity, and feelings of capacity.

Young people I spoke to reflected on how school imparted gendered assumptions regarding ability based on visibility, workplace experience, and the biological categorisation of men and women. For Josh, the lack of women represented in educational materials instigated a recognition of inequality. He said: "In Grade 1, I was starting to learn music and my teacher was showing me videos of people playing music and I thought 'Why are there no women in these videos?'". By minimising women's engagement with music, as an example of representational politics, the teachers at Josh's school reproduced an ideological understanding that women cannot make a career in music. This separation of men and women into different activities is reproduced in labour practices, and the economic conditions, setting up judgements and justifications regarding ideals embedded into culture [80]. Most participants picked up on elements of this uneven structuring of activities and the lack of representation at school and played as developing their sense of inequality.

Recalling their peer group interactions, many young people noted the implication of a gendered order. Participants felt that involvement for boys and girls was emphasised differently with girls often as a side note. These findings show how structural patterns at school separate peers within social groups as a heteronormative practice of exclusion. In these reflections, participants noted gendered expectations where girls felt expression was limited, or in conflict with boys' greater freedom in leisure and play. For some, this meant that they also began to associate greater skill and lower emotional sensitivity with masculine traits and re-evaluate their own potential. Kendra said:

> "I always did engineering and maths and stuff and was always like, "Yeah, I just like to be around the boys because they are less drama". They were not, in fact, less dramatic. They were all awful. But I think that's something you grow up with as well, and it's really hard to unlearn so much internalised misogyny from such a young age ... Yeah, I think it was just that idea of being one of the boys, like this is fine, and it was not fine, and I think the more I grew up, the more I was like, "I do want to be like every other woman, every other woman is really cool. You guys suck."

In this example, Kendra articulates that in adopting tactics to succeed at school where boys were more encouraged and the more emotional qualities of girls her age were dismissed, she realised that boys also engaged in "drama". Another participant, Casey, also spoke of wanting to "be like one of the boys in my small town by playing rough" but was rejected because of their gender. These preconceptions of how to fit in and negotiate play through negating their own expressions show that for some young people, compliance with the dominance of masculinity meant rejecting women and any association with their qualities.

Another way a few of the young women who participated in the focus groups felt that they noticed inequality was the "strong regulation of our physical appearance" at school, as Jessica put it, it was a frequent practice of the school to stop girls' attempts at making their school uniforms more fashionable. This echoes other scholarship about how fashion and uniform regulations for girls in high school are just one of the ways that young women's sexuality is mandated by a masculinist approach to power from an

institutional level [81,82]. Jessica compared her experience at an all-girls school with that of her male-identified friends:

> "I attended a single sex all-girls high school and there was very much a strong regulation of our physical appearance as well. We never had our skirts measured but there was a lot, if certain teachers walked past you, they'd say, you know, pull your skirt down, or make sure you have your hem lowered by this time next week. I had quite a few male friends, or male-identifying friends, who went to single sex male high schools, and they didn't seem to have their appearance as strictly regulated as we were."

Jessica speaks here of realising that the specific restriction of style and expression at her all-girl school was not expected of her male-identified friends attending other schools. Pia also shared experiences where the regulation of uniforms by teachers worked to influence and protect the performance of gender. She said:

> "I guess this is, for a long time I didn't really think I'd be, like, my feelings being at school, and then, like, thinking about what it means to be a woman and the gender equality, I guess weren't really connected; but looking back I connect them. So, when I was in Year 5, so 11, I guess, 11, yeah, 10 or 11 years old, I remember all the girls, this is at a private co-ed school, but all the girls had to kneel in front of a male teacher, or like whoever your teacher was, they would like, measure your uniform with a ruler, which at the time I didn't think anything of but looking back I'm like, that's really creepy."

In this example, Pia reflects on how at the time, she accepted the restriction of stylistic expression, but that this impacted how she felt about being a woman. In doing so, both Pia and Jessica actively problematised how she and her peers had their bodies unfairly objectified and formally disciplined as women. Through this reflection, Pia challenged the vulnerability she experienced as a girl to disrupt the masculine assumptions of power. Outlining the practices of bodily discipline enforced by the school, Pia recognised how this management of her presentation influenced her connection to herself and her gender. Stressing to the focus group that she now realises how "creepy" this practice is, she reflects on the practice of the institution that sexualises the bodies of young women as an insidious and shameful exercise of power.

On the other hand, young men also saw where they and their peers had marked patterns of gender through gendered violence at school. For example, Max said: "I attend, like, a private school in regional Victoria, so I think that coming into that environment was really jarring watching the way that a lot of these kids treated the girls". The men in this study critically reflected on other men's behaviour in maintaining inequality. For Max, the boys in his classes illustrated this behaviour. He said, "I can remember seeing, these guys in my class would push the girls off their chairs as a joke or, like, pull the chairs out from under the girls as a joke and, like, the pack mentality in the boys". Reflecting on growing up in a regional area, Brendan connected with Max about how the explicit became implicit when he relocated to the city. He said:

> "There's a lot more of just 'boys will be boys' mentality. I think that's really the sort of caveat, especially if having gone from regional to living in the city. While it's still fairly prevalent here, definitely not nearly as the same as it is in a regional setting."

In these discussions, Max and Brendan highlight that they have witnessed and perhaps rejected what DeKeseredy and Schwartz [54] describe as patriarchal peer support by describing men's behaviour as 'boys will be boys' or a 'pack mentality'. For these young men, the dominance of men's violence amongst friendship groups built up status, through pitting one against the other and over women. Patriarchal peer support theory highlights how indulging in the dominant and controlling behaviours of boys' friendship groups can lead to dismissing violent behaviour as the norm [54]. As McCook [83] argues, when

so-called typically masculine behaviour is recognised and challenged as these young men have done, it can also be exposed as a point of prevention.

For some participants, it was not until they discussed unequal treatment or experiences with their peers in the later years of high school that they could recognise or come to terms with and challenge those messages about gender inequality. Andrea describes a conversational form of activism emerging between friends as they started questioning the enforced gender roles at school:

> "I think for me, though, it took me a while to really be aware of anything. In primary school I was quite comfortable with the gender roles that were being enforced. I didn't really question it too much until I really got to high school, maybe in Grade Eight, I think, one of my close friends introduced me to the idea of feminism, just through a conversation."

A gradual realisation for many participants enabled them to identify the specific factors that differentiated their experiences of realising, or being able to articulate, what they had felt was gender inequality. While many shared that school could be a place where they saw or experienced the regulation of gender roles and violence (either formal or informal), school also offered opportunities to challenge violence.

Encouragement offered at school can be essential for comfort, solidarity, the verification of practices, and challenging others informally. To illustrate, Anita shared that it was through "opening up" with her friends about shared and distinct experiences of harassment that she realised what happened to her was wrong. She said:

> "It was just more probably discussing with my friends about what has happened and us just opening up, how we have similar experiences and just realising that that isn't right at all. Yeah. I guess a specific example is, I would just see a lot of people, especially guys, at my high school, like, treating people differently depending on their ethnicity, especially like South-East Asian girls and just thinking that that was really creepy and then reading somewhere online they were described as people with yellow fever, and it's kind of like when I realised that this is definitely, it just makes me uncomfortable and that there were other things that also happened in public spaces that also made me feel uncomfortable and could be interpreted as harassment."

As Anita remarks here, during secondary school, the first moments of sexual discrimination and harassment were often compounded with racism. For Anita, it took discussing her experiences with friends to challenge and name what was going on. By bringing up the topic of her harassment with others and understanding what others were going through, Anita verified that what was happening to her was not justifiable. This evidence highlights that exposing the assumptions and the behaviour of others in identifying shared underlying causes in conversations with friends at school can bolster the strength to speak out. Thus, while these participants experienced sexism and harm in high school, these spaces also offered opportunities to strengthen strategies for prevention amongst peers and gave them exposure to feminist ideas.

Some young people in this study had teachers who provided opportunities to include discussions about the extent of gendered violence in class. These classroom conversations meant that those participants were given a chance to challenge their assumptions with evidence and historical examples and to seek guidance. Taylor spoke to the impact of this, saying:

> "During like, our Health PDHPE class in high school, the teacher explained to us that, in fact, that catcalling, even just out on the street, did fit the definition of sexual harassment I knew I was surprised by it, because I thought it was 'normal' and just something that just happens and you just, you know, get on with it. I remember that because it very much challenged what I had assumed up until that point."

These participants highlighted teachers' role in giving young people the strength to challenge ideas that would otherwise be taken for granted or dismissed. Young people saw how their teachers' supported practices of prevention in the structure of classroom content with discussions of gender equity, critical enquiry into the representation within history and revising curriculum, or by supporting an extra-curricular club for peer-led discussions.

Some participants, such as Ruby, said that their tendency to be "a little bit radical anyway" was revealed in the school environment when they noticed that the feminist ideas their parents had shared were relevant to the experiences of their peers. In Ruby's case, this led to sharing ideas and practices with friends at school and forming feminist clubs as an after-school activity:

> "We had a philosophy class-the teacher for that let us sit in the classroom as a club during lunch to discuss consent, gender stuff, all of it. Mum was working for Ansell at the time so she would distribute condoms for the club."

For some, a school-based feminist club offered the chance to enculture solidarity through the re-contextualisation of power through an understanding of their gendered context and changing discursive practices [84,85]. In Ruby's case, her self-created club was a direct response to what she and her peers noted as a missing discourse about gender, sexual ethics, and resources. For these participants, the wider representation of feminism and women as historical figures that challenged assumptions in education and their experiences meant they could reflect on inequality with greater confidence.

### 4.3. "Oh, Everything Makes Sense, It's All Clicked": Finding the Words to Challenge Inequality

While some participants came to terms with these realisations because of their education, many took to social media unguided to explore and discover the language to describe gender equalities. For some, it was not until they had experienced or witnessed the violence that they sought out these resources for themselves. In the following excerpt, Saavi compared her upbringing and lack of access to this knowledge to some of her more class-privileged peers at university. In this example, these young people found resources that they understood with greater ease as they got older when they had more freedom and started to access information for themselves:

> Saavi: It was really interesting, cause I never had that kind of spoken about at school either, but for me something that hit home was when I entered university, a lot of the people I met you know, I went to a public school my whole life and I feel like potentially even socioeconomic status is a factor that plays into how aware you may be about these issues,' cause suddenly all these private school girls were, like, really informed about gender equality and their peer groups at school had been really informed about it, whereas I never had that kind of exposure growing up and I just really wish I did, yeah.

> Grace: I didn't have any equality, I reckon, spoken about at school. For me that would've been very relatively recently. But it's more of a DIY thing like I had to sort of figure it out myself and find my own information, yeah.

> Yas: Yeah, I second that; I relate to that, yeah.

Many of the participants speak of the patterns of gender inequality being an experience they react to in hindsight but could not identify when younger. However, other barriers and contributors to inequality can exacerbate this experience. As reflected in Saavi's discussion with Grace and Yas, the milestones of entering university, having new life experiences, and meeting new people with greater affluence gave them ideas to challenge assumptions. In this reflection, Saavi mentions feeling as though she may have been missing out on "that kind of exposure" or facing further obstructions due to her social class. Noticing something was missing and having to find the information in a self-directed manner, as Grace says, echoes findings from Schuster [63] about how some young people end up looking for and sharing feminist online content.

The visibility of historical figures or popular feminist icons and prominent voices discovered online encouraged the advocacy of some of the participants, such as Lindsey, to "really propel a passion to promote more about education and knowledge". For Grace, it was not until she discovered the words of a prominent author that she began to articulate what she had been feeling, and to explore what she had known to be true:

"An 'Aha!' moment for me. I read *Fight Like a Girl* by Clementine Ford, and this was maybe three years ago, and since then, I think that sort of put all my thoughts onto paper, and it was like reading back my own thoughts and I was like, "Oh my gosh, everything makes sense!" I just couldn't, it was like I couldn't put the words together myself; I knew something was not right with the world, but I just couldn't put the words together. Then it was in book form, and I could just read it back and finally I was like "Oh, everything makes sense—it's all clicked now". Then I can just go and do my own research and advocate how I want to advocate because what I'm thinking is reflected in the book, if that makes sense".

Many participants were encouraged to create change through seeing the words and work of feminist influencers in popular internet culture. However, as Banet-Weiser [86] contends, such economies cannot sustain long-lasting change alone, as they often maintain an unequal relationship with power dynamics that they purport to resist by engaging in misogynistic backlash. This was prevalent in Grace's example of Prominent Australian feminist author and influencer Clementine Ford's bestseller debut book *Fight Like a Girl*, which was published in 2016 and heralded at the time as a significant entry-point for contemporary feminist learning. While many young people in this study appealed to the work of popular feminists who have gained currency in recent decades, they did not seek structural change from their rhetoric and leadership alone.

Most examples of historical inspiration, representation and discourses challenging inequality that were given by the participants explained a context of struggle, or defying expectations, reflecting the participants' own experiences. When asked what messages had challenged their early understandings of gender inequality, participants often referred to examples in the media that were self-representative or relatable. Andrea said that taking the lead from others she saw on social media, she felt the impetus to speak out because of "examples in social media of celebrities speaking out; I started having a bit of ammo in my loaded gun for conversations, I guess. I had something to give to the conversation".

Following the careers of role models through social media also challenged ongoing processes of realisation beyond dominant understandings about feminism. While participants drew inspiration from prominent voices in the media whose advocacy they could follow, they began to see links in the responses those influencers had in their messages. Claire said she saw herself in the actor Emma Watson's character in the popular Harry Potter films and followed her transformation as an advocate online. In her words:

"I think that Emma Watson is a big thing. I think seeing a female character being so necessary to the success of males, seeing her and then seeing her grow beyond that as a person. I followed her, and I have done for years, and I've watched myself evolve along with her as what she posts on social media and the different stuff that she advocates for has changed."

Here, Claire mentions how relating to and following Emma Watson on social media encouraged her growing interest in speaking out against social inequality. For Claire, following a celebrity advocate of a similar age who she related to inspired her to take on her message, and "evolve along with her". Likewise, Meena explains that her admiration of Watson as a role model claiming feminism demonstrated the context of misogyny that she spoke out against. Meena said:

"I think it was still at a time where, like, being called a feminist was a dirty word and stuff like that, and people were like "Emma Watson called herself a feminist; what?". And stuff like that. I'd think about it and be like, "Hey I think I'm a feminist; that's pretty effed up that it's a strange thing to call yourself"."

In this quote, Meena highlights the overriding negative understandings about feminism as a pushback against its goals. Meena's experience of feminist politics in Australia throughout her youth highlighted that despite any sense of feminist 'popularity', this was not always the case. Meena noticed a discord between the acceptance of such role models and broader discourse about feminist campaigning which represented broader structural pressures.

The influence of such icons as entry points for transformative ideas could also create ongoing challenges. For example, while many young people cited the effect of Emma Watson and the Harry Potter franchise, Frances found that they were provoked to become more active upon learning via social media that the author of the book series held transphobic views. They said:

> "J.K. Rowling being openly transphobic to trans women this year. That was a massive thing for me this year, because this was someone I had looked up to for such a long time. When you have to, like, re-evaluate what you think about them, that was a massive learning experience for me."

Here, Frances indicates how significant the influence of a role model can be, including when they counter one's beliefs. In 2019, the author of the Harry Potter franchise, JK Rowling, posted a tweet in support of a woman who espoused views described as 'gender critical'. The following year, Rowling published several tweets and then an essay underscoring her reasons for holding bio-essentialist beliefs about gender, menstruation, violence, and discrimination. These reflections from young people highlight how the rise of feminist icons—even if problematic—often takes place within a context of a postfeminist market that undercuts the goals of equality and changing norms. Some critiques regarding the common white and liberal positionings of feminism rightly point out that its use in discourse is frequently positioned to benefit some and suppress and marginalise others [47,87–89]. Similarly, for young people in this study, it was not enough to be influenced by celebrities to create change, but to seek out voices that were going unheard elsewhere via social media.

Indeed, young people in this study found themselves accessing online archives and resources to question these patterns and seek out the lived experience of others. Asking participants where had seen examples that challenged the gender inequality that they were starting to notice, or if they had begun to look for more information, there was a strong feeling that this was a utility of social media. When I asked participants how they came across such material, two participants shared that they curated their feeds to seek out more information:

> Josh: Yeah, hashtags, reading lists, podcasts, just Googling things and trying to find things that you know, I wouldn't normally have stumbled across.

> Grace: Following like-minded people as well because you know they're going to post content that you relate to, and you want to learn about as well.

Here, participants share that actively seeking out content and discussions about equality online uncovered similar discussions. These practices of unearthing online content highlight that self-directed learning, while a natural process of using social technology, is also driven by need, as participants such as Josh "wouldn't normally stumble across" content about preventing gender-based violence. These actions echo findings discussed by feminist digital activist researchers about the use of social media in response to sexual violence, and the creation of feminist school clubs [47,90].

For some participants, social media enabled them to find like-minded people, and other participants found that social media gave them opportunities to learn about their peers' experiences that they otherwise would not have been aware of. Max recalled:

> "I was probably in Year 7, so around 13. I just remember so distinctly it was on Instagram where I found on an 'explore page' somebody had posted—you know how on Instagram when people will screenshot tweets and other pages and repost tweets. It was something about this girl's experience at high school with this misogynistic principal and I think I just remember being like "Oh my

god, are people like that?", and then I followed that account and found other accounts and it just went on like that."

Here, Max describes how, through the in-built features of Instagram, he became exposed to the experiences of other people his age that had sparked them to call out that behaviour. These experiences recall that while coming across feminist or social justice-related online content can become incidental learning, is often delayed if it is not pursued [47]. In these examples, participants highlight that the affordances of social media for connecting content with similar content, stories and imagery enabled their engagement and helped them develop their understanding and become involved with PVAW campaigns and messaging. Thus, collaborative efforts toward preventing violence must seek to reach and understand how young people are already adapting knowledge and practices to challenge and prevent gender-based violence in their lives.

In the following discussion, Merindah and Andrea talk about how they lacked the skill set or framework to respond to and understand their experiences and that they needed to learn the language that could articulate and share these experiences with others online:

Merindah: I didn't have the language to describe what was happening. I intuitively knew things would make me angry or I knew that this was a clear example of gender inequality. I felt like I didn't have the tool set or the broader systemic understanding of how this fits in with different patterns of my life, and different experiences to then be like, "Oh, well, yeah, broadly this needs to change, then". So, yeah, it was like Tumblr and then Twitter, Instagram, that has given me all the different, I'm going to say theoretical understandings, but has given me the language to, like—

Andrea: [interrupting] I think when you see someone write something so eloquently, that's exactly what I meant, and that happens so often on social media.

Here, Merindah and Andrea discuss the use of specific social media platforms where feminist counter-publics have been organised and distributed to identify and respond to rape culture. For these young women of colour, finding those words from others like themselves affirmed their experiences and enabled their ability to speak out and challenge the cause. Furthermore, the participants of this study found that to find their own practices in response to their rage against inequality or violence, the language that they found online started shifting their attitudes.

As discussing the prevention of gender-based violence became a part of their natural language and use of social media, these young people continued to extend these conversations. These focus group discussions illustrate how social media has informed participants' uptake of ideas and discussions about equality. To illustrate, Oleg said that the use of social media throughout his later high school years encouraged him to equip himself with a stronger vocabulary to speak out about gender inequality:

"It would definitely have been sort of early- to mid-high school on social media and just generally identifying with progressive left views already, and just being, like, "Yes, I agree with that". But I couldn't say specifically, but social media is a part of, like, the language of how, yeah, like, how I've attained some vocabulary around the topic and my idea of whatever activism means."

For these participants, as discussing the prevention of gender-based violence has become a part of their natural language and use of social media, they continue to extend these conversations through these tools. These focus group discussions illustrate the process of how social media has informed participants' uptake of ideas and discussions about equality.

All the same, these discursive practices too often can leave many young people outside the security offered through the gatekeeping and algorithmic mechanisms of social media. For a couple of participants, social media had not always been a place that enabled discussions about gender equality, with Becca saying:

"I don't remember seeing anything on social media about gender-based violence while I was in high school. It was probably when I was a bit older when I was

studying and more so, much more so, in the last few years as I've become more vocal myself and seen more and more on social media because I've been following specific accounts or that kind of thing."

For Becca and others who now work in the family violence sector, it was not until they sought out information after being inspired by their studies that they began to seek more feminist online content and create it themselves. As others suggest, social media frequently offer possibilities for learning and engagement with social justice in private [60]. These findings show how the participants' knowledge to reflect on what they perceived that social media afforded for their own practices, by publicly accessing and sharing how they have been challenged with others to change their behaviour.

The amplification of such content often occurred from to seeking out similar resources and after discussions with friends, seeking to challenge their experiences, or those of others. But as Nat said, "Yeah, for me, I'm only more seeing it now that I'm involved in this work, which means I'm reflecting deeply, but slowly". To her, and other participants–it wasn't until being around others who openly challenged gender-based violence that she became more likely to see such content online. Many had these ideas actively challenged by what they consumed online. Indeed, Paloma said that while she used social media more now, she also understood that social media was not a reliable or comfortable resource to gain greater understanding for many reasons:

"You get Facebook when you're young and you spend all your time on it, and you delve into different corners of it. So, your parents can control a lot of it, especially if you're young but, like, social media kind of just throws it at you and so you can find some really messed up things but things that aren't messed up and more educational as well."

These remarks reflect how the accessibility of feminist discourse online can exist within filtered bubbles out of the same necessity that form counter publics, out of a response to networked misogyny or violence [91,92]. Such understandings in turn situated the participants' knowledge to reflect on what they perceived that social media afforded for their own practices, and the ways in which they engaged with others changed their behaviour.

## 5. Discussion

Materialist feminism challenges the assumption of gendered labour roles under capitalism as essential, natural, or biological [72]. Therefore, the conceptual framework used in this study brings attention to the structures, practices and discourses that shape the understanding of these young people, as they come to acknowledge and challenge gender inequality and gender-based violence in their everyday lives. This approach to social change recognises how moments of discursive intervention can expose and disrupt reproductions of gender inequality. In turn, such interventions can arguably equip young people to recognise, challenge and resist gender inequalities in their own lives.

For the participants in these focus groups, the organisation of gendered relations marked similar effects in several ways, directly and indirectly, to reproduce embedded impressions of inequality. The ways that young people reflect on gender inequalities being noted in their upbringing, schooling, and commented on in the media highlight instances where prevention can be instigated, to support challenging how violence is supported, or started. As Kendra summarised, some participants felt this was part of "growing up as a woman". For many young people in this study, the cognitive effects of violence through gender inequalities were critical to raising both solidarity, and resistance. However, the effect of these practices, i.e., what goes unacknowledged and is accepted, becomes embedded in people's lives through the imposition of power [72,91]. Where many participants spoke of growing up as a girl as shaping how they understood gender inequality, only some of the men indicated an experience of inequality at home or had seen it promoted by others. Many participants located the beginnings of noticing or learning about gender inequality in observations of their family arrangements. Even so, the young people in these focus groups illustrate how providing the support to recognise and challenge

those experiences and the causes of those patterns can influence their engagement with prevention content.

Young people who grew up more directly affected by multiple systems of oppression, such as class, sexuality, and race, spoke about how those power dynamics situated their experience of gender inequality. Notably, while some participants reflected on the imposition of whiteness or class privilege in mainstream media, self-reflection in relation to these dynamics was often missing in our discussions. Western power structures that assume a narrative of the dominant group (e.g., white, middle-class Australians) obscure what might be taken for granted for other participants [77,92]. It is such standpoints that can influence how young people end up approaching and engaging with primary prevention campaign content and activism with the community. Discovering and learning about these ideas through the language and content available online, helped these young people to challenge what they felt went unsaid, or could not put words to themselves.

The examples shared by focus group participants highlight the reproduction of men's dominant social value above women's under capitalism, as has been highlighted by the scholarship on boys' gender strategies in the schoolyard [18,43,93]. Participants related that their capacities to succeed within education settings were set up under heteronormative binaries of competition, with girls being seen as sexualised others to men and seen as being in competition with, or different from, their male peers. For some participants, school friendships also enabled their first moments of dialogue and changing practices in response to gender-based violence, often started by other students. Such affective moments of realisation prompted, for some, what would become more disruptive and resourced practices of consciousness-raising through social technology. Thus, instilling respectful and gender-equal frameworks throughout education settings both formally and informally can promote social cultures that reinforce practices of primary prevention.

Even so, some participants perceived that their backgrounds set them back from engaging with concepts about equality until they were a little older, or through seeing somebody else speak out online. These findings speak to the collective power of peer support and resources available through public engagement and challenges online. Young people in this study found it helpful to see messages that challenged what was otherwise taken for granted or assumed in their culture, or of their experience, by someone influential– as well as other young people. This attention to material difference again highlights how young people use social media to resource themselves with the knowledge and practices that are relevant to them, and others like them [94,95]. Several participants in this study reflected on a missing connection or understanding about gender inequality until they had an affective turn towards engaging in solidarity with others who shared that part of their experience [96–98]. Notably, such comments highlight the importance of making primary prevention content culturally accessible for those who may not otherwise feel resourced to engage with feminism but speak directly to their experiences. For others, this moment of realisation took place due to the words of another, in the form of a book or from following the advocacy of a popular feminist celebrity.

These findings illustrate the value of young people's own perspectives for building strength-based campaigns for primary prevention activities that use the functions of social technology. For such campaigns to benefit their target audience, understanding how and why they are used and interpreted by young people is essential. For them, finding moments that were going otherwise assumed or taken for granted helped them to challenge how instances of harassment, segregation, objectification, and violence could contribute to violence-supportive behaviours with their peers. The young people in this study found that accessing supportive online spaces to reflect collectively upon experiences of gender inequality and those of others enabled them to shift from what was being reproduced. Creating such interruptions online spaces is pivotal to strengthening actions that prevent violence in social settings.

## 6. Conclusions

This paper prioritises the perspectives of young people on the purposes of using social media to share content about challenging gender inequality and preventing gender-based violence. For most participants, early understandings of what to expect came from embodied practices of gender roles in the home and were regulated through school environments. For some, explicitly feminist modes of parenting modelled equal gender frameworks and challenged expectations of violence at home. Others, however, juxtaposed their Western upbringing with a home life that was rich with cultural traditions of hierarchical respect, duty, and honour towards familial roles. School environments promote the socialisation of gender roles, as well as offer a space of influence to challenge the representation of women. However, most participants did not feel a need to expose, challenge or disrupt such messages until they had personally felt their impact, either directly or via someone else.

In this paper, I have discussed how for these young people, gender inequality has been a felt part of their lives that they have had to come to realise, expose or appreciate, or challenge online. For these young people, using social media helped them to find the words for their experiences, or to fill a gap in knowledge from the circulation of or discovery of resources and materials online about gender inequality or violence. Broadly, the existing content and resources were found in a manner that was self-directed to understand and connect with others about their experiences. Engaging with social media as an everyday practice enabled most to articulate their knowledge and to contest previously held ideas from home, upheld in school, amongst peers or within pop culture. For most participants, using social technology was necessary to challenge some existing views, and to understand the lived experience of violence which often is overlooked or dismissed by more public campaign measures to include in discussions about prevention. While much research in the prevention of gender-based violence seeks to engage or intervene on young people to 'create change', these findings indicate how young people themselves are using social technology to do so themselves. The findings of this can strengthen how campaigns direct young people to speak out about violence with others. The factors discussed in this paper show how young people's practices of using social technology give language to preventing gender-based violence in the context of their lives.

**Funding:** This study was possible due to a scholarship through the Australian Government Operating Grant and Research Training Program Scholarship from the RMIT University School of Global Urban and Social Sciences.

**Institutional Review Board Statement:** This study was conducted in accordance with the Declaration of Helsinki, and approved by the DSC College Human Ethics Advisory Network Committee at RMIT University (CHEAN A 21958-01/19 on 15 April 2019).

**Informed Consent Statement:** Written informed consent was obtained from all subjects involved in the study.

**Data Availability Statement:** Data supporting the reported results can be accessed upon request from the author as consent was not obtained to publicise raw transcriptions.

**Acknowledgments:** The author acknowledges her supervisors Anastasia Powell, and Georgina Heydon for the funding acquisition and conceptualisation of this project.

**Conflicts of Interest:** The author declares no conflict of interest.

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
