# Peer review of "“I Didn’t Have the Language”: Young People Learning to Challenge Gender-Based Violence through Consumption of Social Media"

_2673-995X, doi:10.3390/youth2030024_

Round 1
Reviewer 1 Report
This paper tackles an interesting question: the role that social media plays in gender-based violence prevention and has the potential to make an important contribution to the literature.
That being said, I think the paper needs to be re-focused. The title suggests that youth are preventing gender-based violence through social media, which is not the case, they are learning to talk about gender-based violence as a result of their consumption of social media. Second, much of the paper is dedicated to the ways that youth learned about or experienced gender-based violence and only the final part of the paper addresses the role that social media plays in their learning about gender-based violence.
I think the author(s) can reframe the paper in one of two ways: focusing on social media as one avenue by which youth learn about gender-based violence or by centering their use of social media as a source of learning about social media and being challenged by it (e.g. the case of transphobic comments by JK Rowling). As it is currently organized, the paper tells the story of the ways youth experience and/or observe gender-based violence, with only a few examples of the ways in which they use social media to learn more about gender-based violence.
I’m confident that by re-organizing the paper, it will make a contribution to the literature on youth, social media and gender-based violence.
Author Response
Thank you kindly for the considerate comments and suggestions for edits and revisions to give this paper greater focus.
Alongside the edits suggested by R2, I have made the following revisions per your suggestions:
1: The title now reflects the learning and understanding that participants demonstrate in this paper.
2: Adjustment to language throughout the paper indicate that the emphasis on how social media is the tool for learning and putting words to the experiences and reflections of gender inequality and experience described in the earlier findings. This is emphasised with the addition of extra quotes found on pages 19 and 20 in particular.
This focus is illustrated in the language used, but in the discussion and framing of the paper, without losing the emphasis of standpoint reflection given in the findings from the participants - which R2 mentioned was refreshing and beneficial.
Once again, thank you for supporting the work and for your encouraging insights to provide this paper's best reading.
Reviewer 2 Report
I read this article with interest. It is based on a small number of focus groups but makes significant contribution to knowledge given concerns to address gender-based violence from as early an age as possible. Apart from a small number of points which require attention, the article reads very well and offers compelling argument.
Page 2 - I wonder if it would help to give an example or two of 'technological abuse and harassment'. I take this to mean online abuse and sexting, but it would be worth giving illustration of what is meant.
This may just be a journal preference/stylistic thing, but sometimes, as a reader I cannot distinguish between quotations, and words/phrases to which you wish to draw attention or which or ambiguous in some way. See, for example, page 3 'dramatic' or 'reactive'.
Page 6 - there is reference to 'semi-structured' discussions - which is perhaps surprising unless the researcher/s departed from the usual unstructured focus group discussions. A sentence to explain why the focus groups dsicussions were 'semi-structured' and what was meant by that in the context of the research should suffice.
Very interesting to read. The paper brings fresh insights.
Author Response
Thank you very much for these helpful comments and clear direction for revisions.
I have made the following edits on the pages as suggested.
See page 2: a footnote explaining how technology-facilitated abuse broadly encapsulates a wide range of behaviours including image-based abuse, online harassment, nonconsensual sexting behaviour, and so on. References to the work of Powell and Henry (2014) about how young people have been framed in this area is cited, and a more recent national survey on the prevalence of these behaviours in the adult population is given.
Thank you for picking up on this. On page 3, I have made an edit to indicate that this a quotation from literature.
As requested, I have written a sentence on page 6 to indicate that the focus group discussions were semi-structured around key topics to flow the conversation.
Again, thank you for picking up on these issues and your kind words about the paper.
Round 2
Reviewer 1 Report
Congratulations to the author(s) for addressing the concerns raised about the framing of the article. This is significantly stronger and more coherent. I recommend another proof read as there are a few typos, but overall well done!